JDroid: Android malware detection using hybrid opcode feature vector

http://orcid.org/0000-0002-3028-0416 Arslan Recep Sinan sinanarslanemail@gmail.com
Computer Engineering, Kayseri University , Kayseri , Turkey
Arnaiz-González Álvar
Electronic publication date: 2025 Jul 25
Publication date: 2025
Volume: 11
Electronic Location ID: e3051
Received 2024 Nov 22; Accepted 2025 Jun 27
Copyright: © 2025 Arslan
Copyright year: 2025
Copyright holder: Arslan
License: This is an open access article distributed under the terms of the Creative Commons Attribution License, which permits unrestricted use, distribution, reproduction and adaptation in any medium and for any purpose provided that it is properly attributed. For attribution, the original author(s), title, publication source (PeerJ Computer Science) and either DOI or URL of the article must be cited.
License URL: https://creativecommons.org/licenses/by/4.0/

Keywords: Malware detection, Opcode sequences, Hybrid feature vector, Stacked generalized ensemble classifier

Funding: Kayseri University Scientific Research Projects Coordination Unit FKB-2022-1092 This work has been supported by Kayseri University Scientific Research Projects Coordination Unit under grant number FKB-2022-1092. The funders had no role in study design, data collection and analysis, decision to publish, or preparation of the manuscript.

==============================
The rapid proliferation of devices using the Android operating system makes these devices the primary target for malware developers. Researchers are investigating different techniques to protect end users from these attackers. While many of these techniques are successful in detecting malware, they also have some limitations. Because many applications today use advanced obfuscation techniques, advanced disguise, and variant generation techniques to bypass detection tools, this creates difficulties for security experts. However, the rich semantic information hidden in opcodes offers a promising way to distinguish benign applications from malicious ones. In this study, we propose a tool called JDroid that treats opcodes (Dalvik Opcode and Java ByteCode) as features based on static analysis. The proposed tool aims to detect malicious applications with a unique ensemble model in a stacked generalised structure that uses different opcode sequences as a hybrid, and where each feature is first trained separately and then used by an ensemble decision. For this purpose, opcodes are extracted from APK files by code analysis and directly converted into vectors as 0 and 1 according to their usage cases. A subset of 461 features, obtained through filtering and feature selection processes, is then created using fewer features. This increases efficiency and performance, avoids overfitting, and reduces computational cost. The datasets Drebin, Genome, MalDroid2020, CICInvesAndMal2019, and Omer are tested with an application pool consisting of 14 thousand applications, and the classification performance is compared with different machine learning methods. Experimental results show that the proposed approach has an accuracy value of 98.6% and an area under the curve (AUC) value of 99.6% in malware detection without being affected by the obfuscation process.

Introduction

Mobile devices are a practical resource for managing people’s daily needs and communication. They allow us to quickly manage many activities such as banking, social media, education, health, and shopping. In addition, they have additional equipment with sensors such as a camera, GPS, and accelerometer (Arslan, 2021). This popularity, functionality, and proper hardware infrastructure have caused mobile devices to be seriously affected by malicious applications. Cyber hackers want to take advantage of mobile platforms for their malicious activities (showing pop-up unwanted advertisements, accessing important system passwords, stealing confidential information, damaging or using the device in harmful actions, etc.) with various techniques (Khan et al., 2022). Since malicious software is one of the main attack methods used by cyber hackers, it is essential to develop defense systems against this software to control these attacks. Because the data on smart devices is sensitive and private for individuals, it is necessary to detect malware (Muzaffar et al., 2022).

Android has been the primary preferred operating system for mobile devices. Its open-source platform nature (access to source code) has helped many developers and users converge on Android (Bhat & Dutta, 2021; Amin et al., 2020). They are used widely, from the cheapest to the most capable Internet of Things (IoT) devices. According to StatCounter’s statistics for November 2022 (StatCounter Global Stats, 2022), Android operating systems are used in approximately 70.98% of mobile phones. In addition, more than 3M applications were offered to users only on the official application distribution platform in Q3 of 2022. This number is around 1.5M for its closest competitor, the iOS platform. These numbers are much lower in Amazon and the Microsoft Store (Haidros Rahima Manzil & Manohar Naik, 2024). The emergence of cyber hackers has become a nightmare with the widespread use of networks. After all, about 99% of malware produced is for Android (Arslan & Tasyurek, 2022).

Today, the widespread use of mobile devices puts users at risk. To eliminate this risk, signature-based security scans such as Google Bouncer and Google Play Protect are performed. However, despite this, it has not yet been completely prevented. In addition, there are many 3rd party application download platforms for downloading applications on Android, and they are almost non-existent. These applications downloaded from external sources can steal all data stored on mobile devices (images, audio, video, messages, etc.), bank passwords, or demand ransom using them. For these reasons, there is a need for malware detection tools that can protect end users from Android applications. Studies are continuing to be conducted to detect Android malware, which is a problem that has not yet been fully solved.

For malware detection, signature-based detection is used regardless of the operating system (Android, Windows, etc.) (Yu et al., 2018). The encrypted hashes and byte patterns of the files are kept as signature samples of the files. Then, each application is compared with these signature samples, and detection is made. With this method, detecting malicious applications that have not yet been registered in the signature database will not be possible. Google uses the Google Bouncer tool to detect malicious apps on the Android official APK distribution platform, the Play Store (Alazab, 2023). This tool tests the application on a sandbox and is dynamically tested with activity tracking. The application is classified as benign if no malicious activity is found during this process (Chopra et al., 2023). Hackers can escape from this test environment by performing malicious activities after a specific period and then commissioning them after updating. In addition, there are many third-party application environments for Android besides the Play Store, and users can also download from them. Another approach is to assign and classify application features (API calls, code analysis outputs, binary code features) to machine learning techniques. In this method, it will not be possible to analyze obfuscated applications. Exploring applications with the dynamic analysis method is another common approach. It is aimed at classifying applications based on behavior (Arslan, 2022). In this method, with the runtime-related overhead, applications must display their behavior at runtime for analysis to be caught. This situation limits the success of the dynamic analysis method alone (Niu et al., 2020).

Dalvik opcodes (Li et al., 2023) are another type of feature used to predict the behavior of Android applications. Selecting features with the N-gram type of opcodes obtained is preferred (Li et al., 2020; Ali et al., 2020). In most cases, the N value is chosen as three or below, not to increase the number of features. These obtained features are then used to train and test classifiers in machine learning models. However, the limited use of opcodes depending on the N value causes the distinctive features of the application not to be learned sufficiently by the models. To obtain more features, they need to increase the N value. In this case, the number of elements increases rapidly, resulting in a more complex structure for training and classification, requiring more processing time and interfering with some noisy data (Zhang et al., 2018). In deep neural networks (DNN), on the other hand, the training time takes much longer than machine learning models. This causes application analysis to take a long time. Similar situations exist in Java bytecode. However, they are not preferred in the classification of Android applications. In addition, opcodes are the features that provide distinctiveness in obfuscated applications, which is one of the most basic constraints of static analysis (Tang et al., 2022). It is very advantageous in understanding the basic logic of the application.

Preventing risks related to malware and protecting individuals, businesses, or institutions is a dynamic area in terms of cybersecurity and has a very high importance. Our primary motivation in this study is to design a high-performance classifier for Android applications in order to provide this protection task. Our main goal is for this classifier to have a structure that works based on code analysis, can analyze all Android applications, including obfuscated applications, and does this effectively and successfully. Although the obfuscation process changes the code, we think that this change does not cause a logical shift in the SMALI code.

For this purpose, a JDroid tool with a stacking ensemble-based classifier design has been proposed with an original feature engineering approach to develop an Android malware detection tool that can analyze all Android applications, including obfuscated or packed applications, and has high classification performance. Since opcodes are one of the features that can best reflect the working logic of an application, we use them as the main element in feature extraction for classification. JDroid analyzes the usage status of Dalvik opcodes and Java bytecode-based features for each application. Method-specific analyses are obtained using SMALI codes obtained from APK files. In the next stage, a subset of the features obtained as a result of the study is obtained with the ExtraTree Classifier. Then, the classifier designed in the stacking ensemble model structure of the feature vector obtained for each application is used to determine whether the applications are benign or malicious. For the analysis, tests were performed with a dataset containing 14 thousand applications and applications taken from five different datasets.

The main contributions of the article to Android malware detection are as follows: – A model has been proposed that has a malware detection accuracy of 98.69% with the proposed feature engineering and a stacking ensemble model structure, and shows the value of advanced feature engineering and hyperparameter tuning in improving its efficiency.

– Unlike the studies in the literature, an original feature engineering algorithm has been proposed that provides features on SMALI codes of applications based on both Dalvik Opcode and Java Byte Code, and allows it to be used as a hybrid.

– Thanks to feature engineering, code analysis results convert into numerical values, and feature vectors are obtained directly without using any method such as N-gram, context-free grammar (CFG), Image conversion, NLP, or text-based (Word2Vec), textCNN, etc.

– JDroid works and analyzes obfuscated or packed applications without any problems.

– An experimental dataset containing approximately 14 thousand applications, consisting of two different families of malicious and three other benign applications, was prepared. Thus, it has been proven that the proposed model has no data dependency and maintains its high performance independent of the data.

“Related Works” of this article summarizes recent work on Android malware detection. “System Design” explains the proposed methodology, Android application structure, Java, and Dalvik byte code operation code structure. In “Experimental Results”, the results obtained are presented comparatively, and the model’s limitations are mentioned. In addition, comparisons were made with similar studies, and the superior aspects of the proposed model were emphasized. The last part evaluates the study’s results, and information about future studies is given.

Related works

Research has been conducted by many researchers with different perspectives and methods for the detection of malware, and a summary is presented in Table 1. In this study, only OpCode-based static analysis studies were examined. Machine learning based software detection architectures can be categorized as OpCode array, OpCode-based n-gram, API calls, and hexadecimal bytes according to their feature vector. The proposed method in this study stands out among other studies by using different feature vectors as hybrids.

Table 1 Summary of related works based on opcode based Android malware detection.

Article	Year	Features	Dataset (Platform)	Classification algorithm	Reported performance	
Jiang et al. (2019)	2019	Opcode, Sensitive API, STR, actions	Drebin (Android)	KNN, DT, GB	99.5% (AUC)	
Sung et al. (2020)	2020	Opcode sequence as text	BIG 2015 (Windows)	Bi-LSTM, fastText	96.7%	
Zou et al. (2020)	2020	ByteCode	Drebin (Android)	CNN	97%	
Jeon & Moon (2020)	2020	Opcode sequence	Windows hybrid own dataset (Windows)	C-Autoenconder+RNN	96%	
Singh et al. (2020)	2020	Opcode, Permissions, Intents, LSI	CICInvesAndMal2019 (Android)	RF	93.92%	
Niu et al. (2020)	2020	Opcode, CFG	Hybrid own dataset (Android)	LSTM	97%	
Bai et al. (2020)	2020	Opcode, N-gram (N = 5), permissions	Drebin (Android)	CatBoost	96.21%	
Zhang et al. (2019)	2020	Opcode- Bi-gram, API Calls	Hybrid own dataset (Windows)	CNN, BPNN	95%	
Parildi, Hatzinakos & Lawryshyn (2021)	2021	OpCode, Word2Vec	Hybrid own dataset (Windows)	1-Katman: CNN, LSTM	95%	
2-Katman: LR, SVM, KNN, RF, GB	
Bhat & Dutta (2021)	2021	Opcode-Ngram	PRAGuard+Google play store (Android)	Algoritmic solution	96.22%	
Sihag, Vardhan & Singh (2021)	2021	Opcode sequence	Drebin+Genome+Contagio+AndroAutopssy+AndroDracker (Android)	KNN, J48, RF	98.18%	
Darem et al. (2021)	2021	Opcode, N-gram(1, 2, 3), Image conversion	Microsoft malware dataset (Windows)	XGBoost and CNN (Ensemble)	99.12%	
Zhang et al. (2021)	2021	Opcode sequence, System call, Text	Drebin (Android)	CNN-BiLSTM, NLP	97.6%	
Khan et al. (2022)	2022	Opcode, Skip-gram	Own hybrid dataset (Android)	Op2Vec, DNN,	97.47%	
Wang & Qian (2022)	2022	Opcode, Word2Vec	SOREL-20M (Windows)	TextCNN	98.66%	
Jeon et al. (2022)	2022	Opcode sequence, API calls, RGB Images (Hybrid)	HyMalD (Android)	HyMaID (Bi-LSTM, SPP-NET)	92.5%	
Tang et al. (2023)	2023	Bytecode, Hex, N-gram	BIG 2015 and Malimg (Windows)	LightGBM	99.70%	
Xie, Qin & Di (2023)	2023	Dalvik opcode, Permissions, API calls	CIC-AndMal2017 and CICMalDroid2020 (Android)	Stacking Model	96.77%	
Wu et al. (2023)	2023	Opcode, Function-calls, graph2vec	Own hybrid dataset (IOT)	SVM	98.88%	
Lee et al. (2023)	2023	Opcode	Own hybrid dataset (Android)	Multilayer perceptron (MLP)	98.0%	

One of the first studies using opcodes, Karim et al. (2005) provided the generation of models to help classify applications using n-grams and permutations of OpCodes. Tang et al. (2023) developed a malware detection tool based on Bytecode and Hex-n-gram. A total of 99.70% accuracy was achieved with the Unigram and LightGBM classifier. Zou et al. (2020) proposed a model using ByteCode and convolutional neural network (CNN). While the model successfully detects malware, it is also resistant to obfuscation. Bytecodes are primarily used in the literature to analyze Windows-based EXE files. Unlike other studies, the proposed model was used to classify Android applications.

Xie, Qin & Di (2023) proposed a model with the Stacking model classifier, in which other static features are used together with the Dalvik opcode. The model showed 96.77% success in tests with the CICAndMal2017 dataset. Lee et al. (2023) achieved 98.0% success with their models based on opcode categorization and using machine learning models. FAMD (Bai et al., 2020) combines N-grams and permissions derived from Dalvik opcode arrays. They reduced the number of features with their FCBF algorithm and classified them with CatBoost. For N = 5, k-nearest neighbors (KNN) with CatBoost achieved higher performance than random forest (RF) and XgBoost. Zhang et al. (2019) investigates malware detection using a CNN and a back propagation neural network (BPNN) with a hybrid feature vector. Opcode-based features are converted into bi-grams, API calls are converted into vectors according to usage frequency information, and PCA is applied. As a result, 95% success was achieved in the binary classification of applications. CoGramDroid (Bhat & Dutta, 2021) is a model that operates based on OpCode N-grams. Darem et al. (2021) aimed to detect malware with OpCode features, feature engineering, image conversion, and processing techniques. High performance has been achieved with the feature vector consisting of N-gram sequences and an ensemble model.

Wu et al. (2023) achieved a 2% performance increase in malware detection by capturing function calls from opcode sequences and 98.88% accuracy in classification with support vector machine (SVM). Sung et al. (2020) proposed a detection structure based on the fastText model and the Bi-LSTM structure. It uses static opcodes and API calls as features. As a result, it was stated that there was a 1.87% increase in performance compared to similar models. Parildi, Hatzinakos & Lawryshyn (2021) proposed a model in which opcode features are converted to a vector with word2vec in malware detection, and the resulting feature vector is used for classification with a long short-term memory (LSTM) and CNN-based network, and then machine learning (ML)-based algorithms. Zhang et al. (2021) investigated the classification method with convolutional neural network-bidirectional long short-term memory (CNN-BiLSTM) and natural language processing (NLP) techniques using opcode sequences and system calls as text. While the expected success in classification performance was achieved in the study, it was stated that the processing time increased rapidly due to the increase in the number of applications. Sihag, Vardhan & Singh (2021) first obtained the OpCode segments and then converted them into simpler ones. It has been stated that Android malicious application families, application types, and obfuscated applications are successful in their tests with different datasets. Wang & Qian (2022) proposed a model for malicious code classification in which opcode sequences are converted to vectors with Word2Vec and classified with TextCNN. It has been stated that much higher performance can be achieved than with N-gram models, and the constraints in N-gram sequences are avoided. Jeon et al. (2022) proposed a model in which they used static and dynamic analysis approaches together in their work called HyMaID, where OpCode sequence and API calls were used together for feature extraction. For the training of the SPP-Net network, the features produced by the Bi-LSTM were converted into images. In tests for malware detection on Internet of Things (IoT) devices, 92.5% performance was achieved.

Jiang et al. (2019) created a semantic-sensitive opcode sequence using OpCodes, sensitive API calls, and actions. It is aimed at classifying 40 different families. Jeon & Moon (2020) proposed a hybrid model using C-Autoencoder and dynamic recurrent neural network (RNN) networks. It uses static OpCodes as a feature. High performance has been achieved with a shorter processing time. Singh et al. (2020) proposes a model that uses permissions and intents as a hybrid. As a result of the tests with the RF classifier, 93.92% classification success was achieved. Niu et al. (2020) proposed a model in OpCode-based CFG and LSTM structure. While a 97% classification performance was achieved in the tests, it was emphasized that it is a model with low resource consumption.

Sharma et al. (2025) proposed the MOS method, which extracts opcode sequences and splits them into strings. The technique uses filtering and feature selection processes for extracted features. This application, called MOSDroid, has been tested with a dataset of 10,500 applications taken from DataMD, AndroZoo, and Drebin. Experimental results have shown that it has 98.41% accuracy and 99.45% AUC. Gu et al. (2024) proposed an approach to extract features using CodeBERT and TextCNN-based methods instead of extracting statistical information from opcodes and API calls using traditional NLP techniques. Their tool, called GSEDroid, approaches the problem as a graph classification and achieves 99.47% accuracy. Zhou et al. (2024) proposed the FAMCF tool, which detects malware by examining permissions, API calls, and opcodes. In the study on classification with the ensemble model, 92.31% success was achieved for obfuscated applications and 95.32% for unobfuscated applications in the study with the Drebin dataset. Liu et al. (2024), with their tool called SeGDroid, represents API calls graphically with a node representation approach based on word2vec and social network, and performs classification with a CNN-based algorithm. In the tests conducted with CICMal2020, a 98% F-score value was obtained. In addition to the aforementioned studies, there are those that detect malware based on image processing. Kumar & Janet (2022) performed Windows malware analysis using a CNN architecture called DTMIC trained with the ImageNet dataset. The executable files were first converted to grayscale images, and then these images were fed to the deep CNN network. The accuracy of the model was evaluated using the MalImg and Microsoft BIG datasets, with results of 98.92% and 93.19%, respectively. In a subsequent study, Kumar & Janet (2021) proposed a hybrid approach that utilised machine learning and CNN networks for the classification of grayscale images. The experimental results, as measured by the MalImg dataset, yielded an average performance value of 98.28% for machine learning and 98.71% for DL. Kumar (2021) detected Windows malware applications without the need for feature engineering, code analysis, or reverse engineering, employing the MCFT-CNN model. The model was trained using the MalImg dataset and demonstrated 99.18% accuracy and a prediction time of 5.14 ms. In these studies, malware detection is performed without code analysis, but images of each application must be obtained, trained, and tested in a deep network. The method proposed in this study is based on obtaining the smali code from the APK file in a simple way and classifying it with code analysis. Thanks to the proposed static analysis-based practical feature engineering, it is possible to classify using traditional algorithms and achieve high performance. Al-Andoli et al. (2022) proposed a deep learning-based model for malware detection. The proposed methodology constitutes a hybrid approach, integrating backpropagation and particle swarm optimisation algorithms within the deep learning model. A distributed computing architecture was developed to increase the computational efficiency and scalability of the model. The proposed model is tested on standard malware datasets such as Drebin, BodMas, Android_ML, Virusshare_to, and CIC-AndMal-2020. The results show that the model provides high accuracy in malware detection compared to both traditional machine learning (DT, RF, NB, KNN, etc.) and deep learning (CNN, DBN, LSTM) approaches. It also significantly outperformed existing models in terms of computational efficiency and scalability. In other studies by the same authors (Al-Andoli et al., 2023), they stated that deep learning methods are effective in malware detection, and that Gradient Descent optimization and backpropagation algorithms increase the computational cost. As a solution to this, they proposed a model using parallel deep learning classifiers that work on an ensemble basis. Back-propagation and Particle Swarm Optimization models were used as a hybrid for optimization. A parallel computational approach was adopted to increase the efficiency of the computational method of the ensemble model. In this way, both high classification performance was achieved and computations were made 6.75 times faster than traditional methods.

The performances and methods used in the studies given in Table 1, all of which perform analyses using static analysis, vary from each other. Some of these studies (Tang et al., 2023; Zhang et al., 2019; Darem et al., 2021; Wu et al., 2023; Sung et al., 2020; Parildi, Hatzinakos & Lawryshyn, 2021; Wang & Qian, 2022; Jeon & Moon, 2020) perform the analysis of applications prepared for the Windows platform. The structure of the applications ready for the Windows platform is entirely different from the Android application architecture. For this reason, their overall classification performance is lower. The main reason for this is that only opcode sequences are converted to vectors with different natural language processing methods and used for classification. As we suggested in this study, it is not possible to extract and use Java byte codes as features. Instead of feature engineering, classifier designs should be strengthened for increased performance. All studies other than these are for the Android platform, and techniques such as CFG, N-Gram, Word2Vec, Skip-gram have been used to extract features such as OpCode, api, permissions that can be extracted by code analysis of the application and to use these features in trainable models (Khan et al., 2022; Bhat & Dutta, 2021; Tang et al., 2023; Bai et al., 2020; Zhang et al., 2019; Darem et al., 2021; Parildi, Hatzinakos & Lawryshyn, 2021; Wang & Qian, 2022; Liu et al., 2024). In these studies, as many features as possible were obtained with feature engineering, and traditional natural language processing approaches were used to convert them to vectors. This conversion both brings an extra workload and, if the natural language processing method is replaced by large language models (Bertweet, RoberTa, etc.) as it is today, the obtained features need to be converted back to vectors, retested, and evaluated. The analyses made with code analysis have always been used together with a natural language processing method. In addition, there are applications that test the features obtained as a result of the application by converting them into red-green-blue (RGB) images on deep neural networks (Darem et al., 2021; Wu et al., 2023; Jeon et al., 2022; Gu et al., 2024; Liu et al., 2024; Kumar & Janet, 2022, 2021; Kumar, 2021). In these studies, the processing load (conversion to image, deep neural network training, etc.) is relatively high, and their performance is also low compared to the code analysis approach. Since Dalvik OpCodes offer distinctive features in modeling the behavior of Android applications, many studies use them together with machine learning. Some studies use text-based and traditional machine learning (Tang et al., 2022, 2023; Xie, Qin & Di, 2023; Bai et al., 2020; Darem et al., 2021; Parildi, Hatzinakos & Lawryshyn, 2021; Sihag, Vardhan & Singh, 2021; Singh et al., 2020; Sharma et al., 2025; Zhou et al., 2024), while many use statistical models (Khan et al., 2022; Bhat & Dutta, 2021; Tang et al., 2023; Bai et al., 2020; Zhang et al., 2019; Darem et al., 2021; Wu et al., 2023; Parildi, Hatzinakos & Lawryshyn, 2021; Wang & Qian, 2022; Liu et al., 2024). In the studies, they first determine the N value to be N < 3, creating the opcodes’ N-grams. They are then used to train Deep learning networks. At this point, deciding how many N-grams will be sufficient to achieve high performance is necessary. If the N value is selected too small, the classification performance decreases, as many tuples will be produced. If the N value is set too large, there is a rapid increase in processing time and model complexity. When extracting features automatically with DNN networks, it is not easy to understand the effects of features on classification and the relationships between features. In addition, the training time in DNN networks is quite long compared to ML models. In some studies in 2022 and 2023, due to problems in working with N-gram sequences, it was adopted to convert OpCode sequences directly to vectors with the Word2Vec algorithm and to use them in training test-based CNN networks.

In JDroid, unlike the studies given in the literature, both Dalvik OpCode and Java ByteCodes are converted into a fixed-length feature vector and classified with the stacking ensemble model. After the transformation, an easy-to-process feature vector consisting of 1s and 0s is obtained. Thus, N-gram, Skip-gram, Word2vec, Image Conversion+CNN, or directly text-based OpCode analysis is avoided. In addition, Java Byte code was used as a hybrid feature vector with Dalvik OpCode, which has not been tried before in the literature, and higher performance was achieved compared to the Dalvik OpCode. Also, no extra analysis was done for this performance, and all opcodes were extracted from the smali file instead of the classes.dex file. The stacking ensemble model designed for this study was used as a classifier instead of an ML model.

System design

Overview

This study used hybrid feature vector generation and stacked model-based classifier design for Android malware detection. The details of this design are shown in Fig. 1. The methodology mainly consists of dataset collection, preprocessing, feature engineering, and classifier design. In this study, a unique feature engineering approach using a static analysis-based code analysis approach and hybrid feature vector generation and stacked model-based classifier design for Android malware detection has been proposed. The details of this design are shown in Fig. 1. The methodology mainly consists of dataset collection, preprocessing, feature engineering, and classifier design. The first stage of the model, a comprehensive dataset containing three benign and two malicious applications, has been created. Thus, it is aimed to analyze the proposed method without any dependency on the data or dataset. Each application in the dataset is an application file packaged with an APK extension, and it is not possible to perform direct code analysis on these files. At this stage, the application must be opened with reverse engineering methods, and the source code must be accessed. In this study, unlike the studies in the literature, code analysis was performed on SMALI code instead of analysis on Java code. For this purpose, the apktool tool (Arslan, Doğru & Barişçi, 2019) was used. To extract the Java bytecode and Dalvik bytecodes, each application was decompiled using apktool. With the decompilation process, assets, lib, smali, original, res, META-INF, and Androidmanifest.xml files are obtained. The Smali folder contains the files used for the proposed model. Depending on the application design, different numbers of *.smali files are received in the Smali folder. On the Smali codes, which of the 257 different Dalvik OpCodes (invoke-virtual, sub-long/2addr, etc.) were used at least once were scanned and analyzed, and a feature vector was obtained. The feature value was determined as 1 for the used OpCodes, and the numerical value of the feature was defined as 0 for the unused features. Then, the same analysis process was repeated for 204 Java ByteCodes, and their numerical values were set as 1 or 0 according to the usage situation. Thus, two different feature vectors were obtained for the same application. Then, these two feature vectors were combined by concatenation, and a feature vector containing a total of 461 features, the values of which were determined as 0 or 1 for each application, was obtained. Combining both feature vectors has made a hybrid feature vector design with more distinctive features about the application.

Figure 1 The structure of proposed JDroid model and process of predicting.

The hybrid feature vector was used in the training and testing processes of the stacking generalized model. The model is in a two-layer architecture, and the first layer, random forest, extra tree, multilayer perceptron, and gradient boosting classifier are used. The XGB classifier is used as the decision maker in the second layer. The main reason for these choices is that the performance increase cannot be achieved in the ensemble structure with models such as KNN and SVM, which received high results in the first tests. For this reason, an architecture consisting of four models and 1 MLP structure in a tree architecture was designed and tested. Although tree-based models normally perform less than others, much higher accuracy values have been achieved within the designed architecture. The architectural structure was created entirely according to the performance value obtained from preliminary tests. An artificial neural network-based classifier was selected and added to the proposed 2-layer architecture to prevent all classifiers from being tree-based. In addition, in the 2-layer structure proposed in this study, the samples used for training in the 1st layer and those used for testing in the 2nd layer must be completely separated. Because in the testing phase of the 2nd layer, the training examples in the 1st layer are learned. If the training samples in layer 1 are used for testing in layer 2, it will cause overfitting. For this reason, training and test samples are separated from the beginning, as shown in Fig. 1. With complex machine learning models, it’s sometimes easy not to pay enough attention and use the same data at different pipeline steps. This can often lead to good but unrealistic performance or cause strange side effects in others. Therefore, test samples are never used in the training processes of four different classifiers in the 1st layer and the classifier in the decision-making layer in the 2nd layer. In this evaluation, the results obtained when Dalvik OpCode, Java Byte code, and hybrid feature vector are used are evaluated comparatively, and the results are shown in detail in the next section.

Opcode extraction and feature engineering

Dalvik Virtual Machine (DVM) is a virtual environment in which applications coded in Java and packaged with *.apk extension are run for Android operating system devices. Google prefers DVM in the Android operating system instead of the traditional JVM. The reasons for this are (1) being an Apache-licensed virtual machine rather than GPL, (2) the JVM not being suitable for IoT devices due to high resource needs, and (3) the DVM needing less memory and processing time than the JVM. DVM contains processing commands similar to JVM, but needs to be compiled once with the dex compiler after generating the Java byte code. On the other hand, JVM is a virtual environment where software coded in Java can run. It understands Java byte codes and enables them to work on different platforms. Javac is responsible for converting Java code to Java byte codes.

Static analysis refers to extracting and understanding the file containing metadata, source code, and other information within the application package. The Dalvik Executable (DEX) format, the basic format of Android operating systems, includes the code embedded in the Android application file. Different algorithms to parse this compiled code require a linear scan or recursive flows. As a result of this disassembling process, additional files such as assets, libs, smali, res, META-INF, and Manifest.xml are obtained for each Android application. Since the disassembling process requires understanding the compiled code, it involves using the appropriate tool, depending on the architecture. Apktool is one of the most common tools used in this field and was also used in this study. After the disassembled code is obtained, it is possible to use different analysis approaches to analyze this code. These are heuristic approaches: Call Graph, Control-Flow-Graph, N-gram, image-based approach, and code-based modeling methods. This study obtained the *.smali file due to the disassembling process, and a code-based attribute approach was adopted.

In this study, a hybrid feature vector proposal has been introduced with the idea that it can be examined in Java byte codes (Leroy, 2003) together with Dalvik OpCode (Li et al., 2023) in the classification of Android applications created in the Java language and can provide information about the application. Although *.apk files are packaged to run on DVM in the Android operating system, they are in Java bytecode in the previous stage. Moreover, *.smali files produced after decompiling APK files contain Dalvik and Java byte code sequences. Analyzing smali files allows both opcodes to be searched and converted into features. The transaction processes followed in this regard are shown in Fig. 2.

Figure 2 Apk decompilation and proposed hybrid feature vector design of JDroid.

To extract the application features, the smali files of the applications were accessed using the apktool at the first stage. The critical point is that depending on the application’s design, many smali files are in separate folders in the application folder. For this process, software was prepared that automatically scans and combines smali codes and investigates the use of Dalvik opcodes and Java bytecode. Thus, all applications were quickly analyzed, and feature vectors were produced.

In the Dalvik OpCode-based feature vector, a 1 × 257 vector was created for each application, with each of the 257 OpCodes being one if used at least once and zero otherwise. In the Java Bytecode-based feature vector, a 1 × 205 vector was created with the same process. Thus, a fixed-length vector for each application and standardized for classification was obtained. In the last stage, the hybrid feature vector production proposed in this study was performed. The vector contains 461 features for each application.

When other studies in the literature are examined, either of the classes.DEX files have been analyzed to determine the use of Dalvik OpCodes, or only the Dalvik OpCode has been searched in the smali operation code files and used in the classification phase. However, as shown in Fig. 3, Java Byte codes are in the same file. Based on this idea, a hybrid feature vector was created and used for classification. It aims to outperform Dalvik OpCode-based classification by obtaining more application features. At this point, it should be noted that the 1 × 461 hybrid feature vector will require more processing time than Dalvik and Java byte code. To overcome this problem, feature selection has been made instead of evaluating all features, and classification with fewer features has been preferred. Thus, a hybrid feature vector with features that contribute more positively to classification is obtained from both Dalvik OpCode features and Java Bytecode features. As a result, a vector with 256 features for Dalvik OpCode, 204 features for Java Byte code, and 461 features for the Hybrid feature vector is obtained. The primary purpose of this study is to achieve higher performance thanks to more features, and the feature selection process is performed on a huge number of features obtained. Extra Tree classifier is used for this process. Accordingly, the best feature is selected so that the decision tree can be divided according to the Gini index. The feature selection process is performed using this index calculation. In this way, higher malware detection is achieved through fewer and more meaningful features. Accordingly, the best feature is selected so that the decision tree can be divided according to the Gini index. The feature selection process is performed using this index calculation. In this way, higher malware detection is provided thanks to fewer and more meaningful features. There are many feature selection methods available other than ExtraTree, but ExtraTree is quite successful in understanding complex and non-linear relationships between features and classes. In the problem we are addressing, we aim to capture a relationship between the application and the opcodes it uses, and this relationship cannot be linear. However, approaches such as LASSO regression and chi-square assume the existence of a linear relationship, and parameter adjustment is quite sensitive. In addition, ExtraTree is computationally efficient compared to recursive feature elimination (RFE) and, more importantly, it is less affected by noise than the mutual information (MI) feature selection method in high-dimensional datasets. In addition, ExtraTree models have a structure that can be used to embed during training. It instantly ranks and selects features according to importance, so unlike principal component analysis (PCA), the selected features are clear and interpretable. Selected features can be analyzed collectively for malware and benign software. The selected features are directly fed into JDroid’s ensemble model structure.

Figure 3 Dalvik OpCode and Java Byte code usage in same .smali file.

Environmental tools

The Python language, which researchers widely use, was used for hybrid feature vectors and stacking ensemble model design and classification tests. The following libraries were mainly used to develop in the Python language: Numpy: used to manipulate matrices.

Panda is a data processing library; this work is used to process features obtained from Android application files.

Scikit-learn (Fabian et al., 2011): It is an open-source library that hosts machine learning classifiers. This study was used to implement the proposed 12 different classifiers and one stacked ensemble model design.

In addition, Java-based coding was also used to extract approximately 14 thousand applications and automatically generate feature vectors. The experiments used a notebook and CPU (Core i7 Intel 11390H), 32 GB RAM, and a Windows 10 operating system.

Hyperparameter tuning

The proposed model was tested with different classifiers, and the results were compared. GridSearch-CV in the scikit-learn library was used to evaluate TFP and FPR rates automatically and adjust hyperparameters. The CV property was set to 10, and 10-fold cross-validation was performed. For cross-validation, it is necessary to ensure that there are no test examples during training. For this reason, in this study, cross-validation was performed separately for each dataset. In this way, accessing data from the test set during training is prevented, and data leakage is prevented. As a result, the optimal parameters were determined for each classifier and are given below. Logistic regression (LR): C:1.0, kernel:linear, probability=True, gamma:scale

k-nearest neighbors (KNN): n_neighbors=5, p=2, metric=‘minkowski’

Support vector machines (SVM): kernel=‘linear’, C=1.0, random_state=0

Decision tree (C 4.5):

Gaussian naive Bayes (GNB): var_smoothing:1e−9

Multilayer perceptron (MLP): max_iter=1,000, activation=‘relu’, alpha=0.5, hidden_layer_sizes= (10, 20, 10), learning_rate= ‘adaptive’, solver= ‘adam’

AdaBoost (AB): n_estimators:50, learning_rate:0.1

Gradient boosting (GB): loss:‘log_loss’, n_estimators:50, learning_rate:0.1

Random forest (RF): criterion:‘gini’, n_estimators:50, random_state:3

ExtraTree Classifier (ET): criterion= ‘entropy’, max_depth=8

XGBoosting (XGB): learning_rate = 0.01, n_estimators=10, max_depth=5, min_child_weight=1, gamma=0

Linear discriminant analysis (LDA): solver: svd, shrinkage: auto

Experimental results

Dataset design

A heterogeneous and balanced dataset consisting of different datasets was created to be used in this study. Selected datasets have been used in many studies before, and the results have been shared. The number of applications related to the datasets, APK file size, file size after conversion to Java code, file size after conversion to Smali code, and total number of files in Java language were shown in Table 2 in detail. Java ByteCode sequences and Dalvik ByteCode sequences of all applications have been extracted and tested using only Smali files.

Table 2 Dataset list used in test of JDroid.

Dataset	Java ByteCode feature extraction	Dalvik OpCode feature extraction	Type (Malicious or Benign)	Details	
MalDroid2020 (Mahdavifar et al., 2020; Mahdavifar, Alhadidi & Ghorbani, 2021)	✓	✓	Benign	4,039 apps	
Apk: 52.7 GB	
Java: 307 GB	
Smali: 184 GB	
Files (analyzed): 29,684,927	
CICInvesAndMal2019 (Keyes et al., 2021; Rahali et al., 2020)	✓	✓	Benign	1,648 apps	
Apk: 20.06 GB	
Java: 109 GB	
Smali: 65.2 GB	
Files (analyzed): 10,410,609	
Omer (Doğru & Kiraz, 2018)	✓	✓	Benign	1,346 apps	
Apk: 20.5 GB	
Java: 90.8 GB	
Smali: 45.7 GB	
Files (analyzed): 7,460,093	
Genome (Zhou & Jiang, 2012)	✓	✓	Malware	1,239 apps	
Apk: 1.51 GB	
Java: 4.94 GB	
Smali: 2.56 GB	
Files (analyzed): 427,871	
Drebin (Arp et al., 2014)	✓	✓	Malware	5,541 apps	
Apk: 6.79 GB	
Java: 24.6 GB	
Smali: 13.86 GB	
Files (analyzed): 1,959,954	
Total	✓	✓	Malware + Benign	7,025 Benign, 6,774 Malware	
Total: 13,799	

Beneficial applications refer to reliable applications that are made available to meet user requirements. In this study, MalDroid2020, CICInvesAndMal2019, and Omer, which have been used and validated in various studies, were used. As given in Table 2, there are a total of 8,176 applications in these datasets. The main problem with applications is that they can have similar Dalvik bytecode and Java bytecode calls. This may cause the training and test samples to be the same and cause overfitting. Other apps may have the same feature sets. For this reason, the same ones were eliminated. When the data was examined, it was determined that 1,151 of them were the same and repetitive applications were removed from the dataset, and as a result, a benign dataset containing 7,025 APKs was created. Since there was no severe sample selection imbalance during the random selection of the training and test sets, stratified sampling was not used. Because when randomly selected, the number of samples for training and test samples is approximately the same.

While malicious apps meet user requirements, they mainly aim to harm users. Often, these purposes are hidden in a part of the application. Ways of harm can be executing malicious program code without the user’s knowledge, repeated execution, or lingering on the device for longer, making it difficult to detect. This study used 6,774 of the 6,780 applications in the Drebin and Genome datasets.

The dataset, compiled from multiple malware datasets and comprising approximately 13,799 APKs (7,025 Benign and 6,774 Malware), was randomly divided into two subsets for training and testing the models. In this context, 70% of the data was allocated to the training set to facilitate the model’s learning process. In comparison, the remaining 30% was assigned to the test set to assess the model’s accuracy and its ability to generalise.

Malware detection with Java feature vector, Dalvik feature vector and Hybrid vector for all classifiers

This study investigated the effect of hybrid feature vector design on Android malware detection. Unlike other studies, using both Dalvik OpCode and Java Byte code in the same file was analyzed and used for classification.

To evaluate the classification performance of the proposed model, the test results of only Java Bytecode, only Dalvik OpCode, and the hybrid feature vector in the same classifier were taken comparatively and are given in Table 3. Tests were repeated with 13 different algorithms in different classifier types, and the results were measured according to accuracy, precision, recall, and F-score metrics. The Hybrid feature vector and the Stacked Generalized ensemble model achieved the highest success. In addition, the hybrid feature vector design performed better in all classifiers than the Java and Dalvik-based feature vectors. In addition, with the hybrid feature vector, a 1.5% performance increase was achieved in some classifiers, while it was 1% in others.

Table 3 Results of all the experimented models (Java bytecode, Dalvik OpCode and JDroid hybrid).

Type	Classification algorithm	Java byte code	Dalvik OpCode	JDroid (Hybrid FV)	
Acc	Pre	Acc	Pre	Acc	Pre	
Rec	F-score	Rec	F-score	Rec	F-score	
Regression algorithm	LR	92.9	90.6	95.9	95.2	96.6	95.9	
95.1	92.8	96.2	95.7	97.1	96.5	
Instance-based algorithms	KNN	95.3	94.5	97.7	97.1	98.1	97.3	
95.8	95.1	98.2	97.6	98.8	98.1	
SVM	95.6	94.2	97.3	95.7	98.1	96.8	
96.8	95.5	98.8	97.3	99.2	98.0	
Decision tree	C 4.5	95.4	94.5	97.3	96.5	97.7	96.9	
95.9	95.2	97.9	97.2	98.3	97.6	
Bayesian	GNB	90.7	86.6	93.0	92.0	94.3	92.6	
95.4	90.8	93.7	92.8	95.7	94.2	
Artificial neural network	MLP	95.0	94.1	97.3	96.5	97.6	97.2	
95.6	94.9	97.8	97.2	97.9	97.5	
Ensemble	Ada boosting	93.0	90.5	95.4	94.9	96.5	95.9	
95.5	92.9	95.6	95.2	96.8	96.3	
GB	94.3	92.9	97.3	95.8	97.7	96.6	
95.5	94.2	98.7	97.3	98.6	97.6	
Stacked generalization (L0: RF, ET, MLP, GB; L1: XGB)	95.7	95.1	98.1	97.2	98.6	97.9	
96.0	95.5	98.9	98.1	99.2	98.6	
RF	93.9	91.4	96.9	95.2	97.3	95.8	
96.3	93.8	98.4	96.8	98.7	97.2	
ET	94.5	92.8	97.4	95.6	98.1	96.7	
96.0	94.4	99.0	97.3	99.4	98.0	
XGB	92.9	90.5	95.8	95.2	96.7	95.9	
95.3	92.8	96.1	95.7	97.2	96.6	
LDA	LDA	92.7	91.1	95.9	94.7	96.2	95.2	
94.0	92.5	96.8	95.7	97.0	96.1	

As a result, the hybrid feature vector, created by taking the most meaningful features for Java Byte code and Dalvik OpCode, showed higher classification success despite using a similar number of features, thanks to feature selection.

A cross-validation (CV) process was carried out to solve overfitting, bias, and generalization problems regarding the results obtained in this study. Thus, it was intended to reduce bias and demonstrate data independence. As it is known, CV provides the ability to predict performance on data that is not used and seen during training. In addition, another problem that CV solves is to prevent sampling instability. Randomization due to a single selection of training and test data may cause experiments to produce unstable results in the case of a different division. Cross-validation tried to demonstrate overfitting control, bias reduction, and data independence by evaluating all these situations together. Cross-validation graphs obtained with the model proposed are given in Fig. 4.

Figure 4 Cross-validation scores for all classifiers for three opcode vector representation.

(A) Dalvik Opcode (B) Java Bytecode (C) Hybrid Feature Vector (JDroid).

As can be seen in graph A, high classification performance can be achieved for all classifiers except GNB, and the results are similar in repeated tests. The highest classification success was obtained with the stacking ensemble classifier proposed in this study. Graph B shows the classifier’s results based on Java bytecode. As can be seen in the results, a lower classification performance is obtained than the Dalvik OpCode. As can be seen in the (C) graph, the highest classification success is obtained with the hybrid feature vector, independent of the classifier, since only the significant feature is taken. These results show that the proposed hybrid feature vector and stacking ensemble model design contribute positively to the classification.

The ROC curve for all classifiers is taken and presented in Fig. 5. Accordingly, in parallel with the evaluations made according to the Accuracy value, the highest value was obtained with the hybrid feature vector and stacking classifier, with a rate of 99.6% in the area under the ROC curve (AUC) measurements. The results confirm each other and demonstrate the success of the proposed model.

Figure 5 RoC curve for all classifiers for three opcode vector representation.

(A) Dalvik Opcode (B) Java Bytecode (C) Hybrid Feature Vector (JDroid).

Detection of malware with proposed stacked ensemble model

Accuracy and AUC measurements do not allow for the evaluation of the results on a class basis. Three different feature vectors were tested with the stacking ensemble classifier, and the confusion matrices are given in Fig. 6. The hybrid feature vector captured the lowest FP and FN values, as shown in Fig. 6C. Only 57 false detections were made in 4,140 test samples, and the classification performance was 98.6%. In addition, it has been observed that malicious applications are classified with higher success than benign applications. While there were only 13 false detections in malicious applications, 44 false detections were made in benevolent applications. For this reason, while the recall value increased to 99.2%, it was above the accuracy value, while the precision value decreased to 97.9%.

Figure 6 Confusion matrix for three feature vector and stacked ensemble model.

(A) Dalvik Opcode (B) Java Bytecode (C) Hybrid Feature Vector (JDroid).

The different test results obtained in this section support each other, and it can be seen that the FP and FN values of the hybrid feature vector and stacking ensemble classifier are low. In addition, due to the hybrid feature vector having more features, it can be thought that there will be an increase in training and testing times, which is one of the disadvantageous points of the proposed model. Figure 7 shows the measurement results of training and test times. Measurements were made for all three feature vectors with the stacking ensemble classifier, where the highest performance was achieved in the tests.

Figure 7 Training and testing time graph for three feature vector design.

There was no significant change in training times for all three feature vectors. The measurement is shown proportionally, not in ms. Thus, the effect of the computer infrastructure change was eliminated. Accordingly, during the training period, the Java byte code requires approximately 20% less processing time than the Dalvik OpCode and Hybrid feature design. The test obtained balanced results regarding processing time in three different feature vectors. The most important reason for this is the use of a similar number of features in all three vectors due to feature selection.

Discussion and limitations

Android app developers use various techniques to prevent malware from being caught by detection tools or reduce their code’s readability. Because with multiple devices such as apktool and dex2jar, there is a chance to open almost all applications with reverse engineering methods and access their source codes. However, it is much more challenging to analyze the states of applications packaged in an obfuscated form and make them easy to read. The limitations of the proposed model in terms of such cases are evaluated in this section.

The obfuscation process refers to how some transformations (renaming, junk code injection, control flow-based obfuscation, etc.) are made by preserving the operation and meaning of the source code. Different codes emerge each time the same source code is repeatedly obfuscated. However, capturing some similarities within these other codes is beneficial in terms of classifying the application. In most cases, changing opcodes or bytecodes is impossible, while manual code analysis is prevented. The transaction code is needed if any action is to be taken on the system.

The model proposed in this study can resist various obfuscation techniques such as name changing, junk code injection, and control flow-based obfuscation. Especially in cases of junk code injection, the operation code search time may be prolonged, and negative situations may arise regarding operation time. This will be seen in the details section of the dataset table. The size of smali file sizes is critical. This problem was solved by terminating the search process once the relevant transaction code was reached. However, this may cause problems for some applications. In addition, the inherent limitations of machine learning models apply to the model proposed in this study. If real-time data is fed to the proposed model, the model cannot continuously train. The most basic way to overcome this will be to periodically include new applications in the dataset and update the proposed model by retraining. Therefore, further research is needed in the future.

In this study, as given in Table 4, more than one malware dataset was brought together heterogeneously to obtain a dataset containing sufficient benign and malicious applications. This use means that training and testing processes are carried out based on the assumption that the applications are equivalent in all datasets. Sensitivity analyses may need to be performed depending on the datasets’ weight. Whichever database is more dominant, the training of the data may be more dependent on it. To overcome this limitation, a dataset with larger samples and sufficient numbers for both benign and malicious applications will be created.

Table 4 Comparison of classification results with proposed JDroid model.

Article	Year	Features	Dataset (Platform)	Classification algorithm	Reported performance	
Jiang et al. (2019)	2019	Opcode, Sensitive API, STR, actions	Drebin (Android)	KNN, DT, GB	99.5% (AUC)	
Zou et al. (2020)	2020	ByteCode	Drebin (Android)	CNN	97%	
Sihag, Vardhan & Singh (2021)	2021	Opcode sequence	Drebin+Genome+Contagio+AndroAutopssy+AndroDracker (Android)	KNN, J48, RF	98.18%	
Darem et al. (2021)	2021	Opcode, N-gram (1, 2, 3), Image conversion	Microsoft malware dataset (Windows)	XGBoost and CNN (Ensemble)	99.12%	
Zhang et al. (2021)	2021	Opcode Sequence, System Call, Text	Drebin (Android)	CNN-BiLSTM, NLP	97.6%	
Khan et al. (2022)	2022	Opcode, Skip-gram	Own hybrid dataset (Android)	Op2Vec, DNN,	97.47%	
Wang & Qian (2022)	2022	Opcode, Word2Vec	SOREL-20M (Windows)	TextCNN	98.66%	
Jeon et al. (2022)	2022	Opcode sequence, API calls, RGB images (Hybrid)	HyMalD (Android)	HyMaID (Bi-LSTM, SPP-NET)	92.5%	
Tang et al. (2023)	2023	Bytecode, Hex, N-gram	BIG 2015 and Malimg (Windows)	LightGBM	99.70%	
Xie, Qin & Di (2023)	2023	Dalvik opcode, Permissions, API calls	CIC-AndMal2017 and CICMalDroid2020 (Android)	Stacking model	96.77%	
Lee et al. (2023)	2023	Opcode	Own hybrid dataset (Android)	MLP	98.0%	
Wu et al. (2023)	2023	Opcode, Function-calls, graph2vec	Own hybrid dataset (IOT)	SVM	98.88%	
Sharma et al. (2025)	2025	Opcode	DataMD, AndroZoo, Drebin	DNN	97.91%	
Gu et al. (2024)	2024	Opcode CodeBERT, textCNN,	Drebin, Androzoo	GNN	99.47%	
Zhou et al. (2024)	2024	Opcode, permissions, api calls	Drebin, CICInvesAndMal2019	Ensemble	95.32%	
Liu et al. (2024)	2024	Api calls Word2vec,	CICMal2020	GNN	98%	
JDroid (Proposed Model)	2024	Java ByteCode and Dalvik Opcode Hybrid Feature Vector- Fixed length array (without N-gram, Skip-gram, Word2Vec, Image Conversion, Text)	Hybrid Dataset (Android)	Stacking generalized ensemble model	Acc: 98.69%	
AUC: 99.6%	

In the evaluation of the computational complexity of the proposed model, the utilisation of Java byte code and Dalvik OpCode features as a hybrid approach is employed. This means that more features are extracted from an APK file. This is a factor that increases the computational complexity of the proposed model. However, in this study, unlike other static analysis methods such as N-gram and API-Call, in the feature extraction phase, the research is analyzed on “smali” files (instead of Java codes) over a limited OpCode list (461 different OpCodes), and only the existence of the relevant OpCodes is investigated. For this reason, it is not necessary to scan the entire file, and the research is terminated when the relevant OpCode usage is encountered. All these operations provide a generalizable method in all applications, and it is possible to complete the research quickly. In addition, since researching the use of API calls over the smali code requires deep analysis (filtering the instructions starting with “invoke”, etc.), the processing time is quite long. When the analysis is done on Java code, the computational cost increases, and more importantly, it is quite problematic to do this research on obfuscated Java code. In the N-gram algorithm, the feature dimensionality tends to increase exponentially rapidly depending on the n value. The necessity of N-gram shifting and the extraction of the related feature vectors is a situation that increases the computational complexity. As a result, the model proposed in this study, which works on an opcode basis, is an approach that increases the computational complexity due to the use of hybrid feature vectors, and it is also in an advantageous position compared to other static analysis methods, such as N-gram and API call analysis. In order to further increase the computational efficiency and scalability of the proposed model, it is possible to use a distributed parallel computing architecture as in the studies (Al-Andoli et al., 2022, 2023, 2024). This situation will be especially taken into consideration in our subsequent studies.

JDroid works based on static analysis. Consequently, it does not require any sandbox, etc., operating environment as in dynamic analysis. This makes JDroid integratable to mobile antivirus programs or leading target enterprise application distribution platforms (Google Play Protect, etc.) or mobile antivirus solutions (Bitdefender Mobile, etc.). For example, during the application installation phase, the Java byte code and Dalvik OpCodes of the application are extracted, and the malicious status of the application can be analyzed by testing it with the previously trained model. In addition, investigating both Java byte code and Dalvik OpCodes on mobile devices and analyzing their usage will bring additional computational cost due to limited resources. In order to reduce this, JDroid works with smali files instead of using the source code, which consists of Java files. Then, the features are reduced before classification with feature selection algorithms on the features. In addition, the computational cost can be reduced by optimizing the feature extraction methods or by methods such as distributed computing (Al-Andoli et al., 2022, 2023, 2024). This will allow JDroid to be used for real-time analysis. The main distribution target of JDroid is enterprise companies that provide application market management and expect high-accuracy detection, and allow cloud-based analysis. However, a version that performs fast detection and works with fewer attributes can also be used on end-user Android devices, thanks to the structures that can be connected to a full-featured system in the cloud for in-depth analysis when needed. In the future, we plan to develop a prototype to evaluate the efficiency and scalability of JDroid on Android platforms.

The use of Grid-SearchCV for hyperparameter selection cannot always guarantee the best parameter selection because the search is performed for a specified range. For this reason, it cannot be guaranteed that the results obtained are always the best value.

Comparison of similar works

The problem to be solved in this study has been studied for many years, and a comprehensive review is given in the literature summary section. High-performance studies are presented in Table 4, compared with the JDroid application, and their advantages and disadvantages compared to other studies are highlighted.

When the studies are evaluated in general, it is understood that Android malware detection is done by feeding different classifiers with code-based features such as OpCode, api calls, permissions, and bytecode based on static analysis. In our study, a similar opcode-based feature extraction process was carried out. However, in all studies in the literature, opcodes were investigated by considering the Dalvik OpCode list. In this study, however, research was conducted on the same code together with the Dalvik OpCode list and also in the Java Bytecode list. Although both are expressed as OpCodes, they are two completely different code structures. Our main purpose in doing this is to obtain more features about the application. As can be seen in the studies in the literature, it is understood that more meaningful features about the application are intended to be obtained by extracting different features, such as api calls, permissions, together with the opcode in many studies. In this study, instead of looking at api calls and permissions, we considered Java byte codes. In addition, in most of the studies, different statistical or LLM-based feature transformation algorithms such as word2vec, codebert, textcnn, graph2vec, n-gram, and skip-gram were used in the feature analysis phase. Because obtaining OpCode sequences is not enough for direct classification, feature vectors need to be obtained from these features. In this study, without using such a method, only the vector conversion process was performed in a way that would be one if the relevant OpCode is used by the application and zero if it is not used. Thus, feature vectors were obtained quite effectively. As a result, for JDroid, there was no need for complex structures such as N-gram, CFG, image conversion, or text-based analysis.

When evaluated in terms of tested datasets, there are studies in the literature that use specific datasets, as well as applications that test with 10 thousand or more application pools. In this study, tests were performed with a very high number of application pools, such as 14 thousand. Therefore, it was shown that the JDroid tool does not have any dependency on any dataset.

In the selection of classifiers, it is understood that in recent years, either CNN-based GNN or DNN architectures have been used. For CNN-based architectures, OpCode sequences need to be converted to images or graphics with different methods, which means extra workload. In this study, a traditional machine learning-based ensemble model is used, and no image transformation, etc., is required. After scanning the OpCode sequences, feature vectors are obtained directly. In terms of performance, the high AUC and Accuracy values obtained by JDroid are better than many studies in the literature.

Conclusion

Today, many researchers are investigating how to stop the rapid growth of Android malware and how to detect it with an effective and efficient classifier. In this study, JDroid, an innovative method that is not affected by obfuscation and uses OpCode sequences for Android malware detection, is proposed. JDroid treats Dalvik OpCodes and Java bytecode sequences as a hybrid and classifies them with an ensemble model. Considering two different OpCode sequences as a hybrid allows the distinguishing aspects of the features from both methods to be used effectively and provides more semantic features about the application. The effectiveness of the model was evaluated by testing with an application pool containing Drebin, Genome, MalDroid2020, CICInvesAndMal2019, and Omer datasets, and 14 thousand applications. Thus, it was shown that the proposed model is robust against applications from different application datasets and has no data dependency. As a result of many comparative tests and analyses carried out by taking into account the factors affecting the model performance, 98.6% accuracy and 99.6% AUC values obtained confirm the effectiveness of JDroid and provide a solid start for future studies.

In the future, it is planned to work on the constraints given in the 4.4 limitations section. The focus will be on optimizing results and classifier design—the tests will be performed with a larger dataset containing more applications. In addition, with the increase in the number of examples in problem-solving, studies will be carried out on classification with deep learning models. Additionally, designing more complex and technical models with multiple layers that work on a divide-and-conquer basis will be tried in the future. Thus, while higher performances will be achieved, more efficient models will be created.

Supplemental Information

Supplemental Information 1 Dataset.

Supplemental Information 2 README.

Supplemental Information 3 Source Code.

Supplemental Information 4 Raw experimental data.

Supplemental Information 5 List of datasets used in this article.

A native English speaker was employed to check the grammar of the study. An earlier version of this manuscript was checked for grammatical errors using Grammarly.

Additional Information and Declarations

Competing Interests

The authors declare that they have no competing interests.

Author Contributions

Recep Sinan Arslan conceived and designed the experiments, performed the experiments, analyzed the data, performed the computation work, prepared figures and/or tables, authored or reviewed drafts of the article, and approved the final draft.

Data Availability

The following information was supplied regarding data availability:

The source code is available in the Supplemental File.

The third-party datasets are available at:

- MalDroid2020 https://www.unb.ca/cic/datasets/maldroid-2020.html.

- CICInvesAndMal2019 https://www.kaggle.com/datasets/malikbaqi12/cic-invesandmal2019-dataset.

- Omer: https://iotlab.gazi.edu.tr/ available via email to Prof. Dr. İbrahim Alper DOĞRU at iotlabgazi@gmail.com.

- Genome: http://www.malgenomeproject.org/ available via email to Yajin Zhou at yajin_zhou@zju.edu.cn.

- Drebin: https://drebin.mlsec.org/ available via email to Daniel Arp, Michael Spreitzenbarth, Malte Hübner, Hugo Gascon, and Konrad Rieck at drebin@mlsec.org.

The raw data is available in the Supplemental File.

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
