# Peer review of "JDroid: Android malware detection using hybrid opcode feature vector"

_PeerJ Computer Science, doi:10.7717/peerj-cs.3051_

## Round 0.1 · original submission · Major Revisions

Please take into account carefully the reviewers' suggestions, specially those related to the experimentation and the explanation of the proposed method.

Reviewer 1 ·

Basic reporting

No comment

Experimental design

The experimental scenario are very nicely articulaed.

Validity of the findings

Good work

Additional comments

Annotated reviews are not available for download in order to protect the identity of reviewers who chose to remain anonymous.
Cite this review as

Reviewer 2 ·

Basic reporting

Lack of clarity in sentences and most of the manuscript is written ambiguously. Authors are suggested to use clear and unambiguous, professional english in their manuscript.

Experimental design

Authors have not clearly explained thier methodology and feature engineering in Section 3.

Validity of the findings

The proposed methodology itself more ambiguous. So the results cannot be assessed.

Additional comments

Authors have addressed clear contributioons in introduction section. Literature review effectively conducted and given a good summary of related works. But authors have failed to explain the proposed method, feature engineering and feature selection in their work.

Cite this review as

Reviewer 3 ·

Basic reporting

The authors developed JDroid, an Android malware detection approach using a hybrid opcode feature vector. The work is promising but requires some revisions:
1- The introduction section needs to be improved, particularly by emphasizing the motivation and novelty of the study. The current introduction does not clearly highlight why this research is important or how it advances the field of Android malware detection.

2- The related work section can be extended to include recent techniques for malware detection.

3- The paper does not sufficiently elaborate on how the combination of Dalvik opcode and Java Byte code in the hybrid feature vector improves detection performance, particularly in handling obfuscated or polymorphic malware.

4- The results section would benefit from the addition of statistical analysis to demonstrate the significance of the findings. Currently, the results are presented without sufficient statistical validation.

5- The conclusion section should be revised to better summarize the key contributions of the study and highlight its potential impact.

Experimental design

Overall okay. 3- The paper does not sufficiently elaborate on how the combination of Dalvik opcode and Java Byte code in the hybrid feature vector improves detection performance, particularly in handling obfuscated or polymorphic malware.

Validity of the findings

The results section would benefit from the addition of statistical analysis to demonstrate the significance of the findings. Currently, the results are presented without sufficient statistical validation

Cite this review as

---

## Round 0.2 · Major Revisions

Please take carefully into account the reviewers comments before the next submission. In case the reviewers consider major decision after the next round of revision, the paper could be rejected.

Reviewer 3 ·

Basic reporting

The paper "JDroid: Android Malware Detection Using Hybrid Opcode Feature Vector" discusses an interesting and relevant topic in Android security. The proposed hybrid feature vector approach is novel and addresses key challenges in malware detection. However, the paper requires some improvements before being considered for publication.

Main Concerns:

Computational Cost Analysis – While the hybrid feature vector improves accuracy, it may also increase computational complexity. A runtime analysis comparing JDroid with traditional methods (e.g., N-gram, API-call-based detection) would be beneficial to assess its efficiency.

Related Work Expansion – The related work section is comprehensive but could benefit from more discussion on recent advancements in malware detection methods. The authors are encouraged to consider and discuss the following relevant studies:
DOI: 10.1016/j.asoc.2022.109756
DOI: 10.1109/ACCESS.2023.3296789
DOI: 10.1109/ACCESS.2024.3354699

Practical Deployment Considerations – It is recommended to discuss how JDroid could be deployed in real-world malware detection systems. Addressing integration challenges, real-time detection feasibility, and potential deployment strategies would enhance the practical impact of the study.

Experimental design

Comparison with Deep Learning Models – The study lacks a direct comparison with deep learning-based malware detection methods (e.g., CNN-based classifiers). Including such a benchmark would provide a more comprehensive evaluation of JDroid’s performance against state-of-the-art methods.

Feature Selection Justification – The authors use ExtraTree for feature selection but do not justify why it was preferred over other methods like mutual information or LASSO regression. A brief comparison of feature selection techniques would improve clarity.

Validity of the findings

Dataset Bias and Cross-Validation – The paper should explicitly state if stratified sampling was used to balance malware families. Additionally, clarification is needed on whether cross-validation was performed across datasets or within each dataset separately to ensure fairness and avoid data leakage.

Additional comments

Language and Clarity – The manuscript is generally well-written but contains minor grammatical errors and awkward sentence structures. A thorough language revision is recommended to improve readability and clarity.

Cite this review as

---

## Round 0.3 · Minor Revisions

Dear authors,

The manuscript has improved significantly since the first version, nevertheless there are still a couple of problems related with the experimentation:
- It is neither complete nor well-presented: the source code must be uploaded to a public repository (Github for example) with a README that contains all the instructions to be reproducible.
- The source code uploaded to the platform is not enough for running the experiments.

Take this into account, science must be public and reproducible.

Moreover, read again the manuscript since there are still some minor typos in it.

Regards,

Reviewer 1 ·

Basic reporting

Clear and unambiguous contents and grammar is used throughout the manuscript.

Experimental design

Yes

Validity of the findings

All underlying data have been provided; they are robust, statistically sound and controlled.

Additional comments

Recheck on flow and English grammar mistakes.

Cite this review as

Reviewer 3 ·

Basic reporting

The authors have adequately addressed my concerns. The manuscript is suitable for publication pending minor polishing of the writing.

Experimental design

No further comments

Validity of the findings

No further comments

Cite this review as

---

## Round 0.4 · Minor Revisions

There are inconsistencies between the source code, the README file and the description on the paper. Please, check carefully whether CV or train/split is used, hyperparameter tuning, etc.

Moreover, the upload of the source code to a public repository is advisable.

---

## Round 0.5 · Minor Revisions

Thank you very much for all the information on the answer document. Nevertheless, the train/test split (70/30) that is mentioned there is not properly explained in the manuscript.

Please, take into account that the experimental setup of the paper must be as accurate as possible and in the current version does not fit with the explanation of the answer document.

---

## Round 0.6 · accepted · Accept

All the details have been corrected in the current version of the manuscript.